

# Adverse events associated with incretin-based drugs in Japanese spontaneous reports: a mixed effects logistic regression model

Daichi Narushima[1], Yohei Kawasaki[1], Shoji Takamatsu[2] and Hiroshi Yamada[1]

[1] Drug Evaluation & Informatics, University of Shizuoka, Shizuoka, Shizuoka, Japan
[2] Office of Safety II, Pharmaceuticals and Medical Devices Agency, Tokyo, Japan

Corresponding author
Hiroshi Yamada,
hyamada@u-shizuoka-ken.ac.jp

## ABSTRACT

**Background:** Spontaneous Reporting Systems (SRSs) are passive systems composed of reports of suspected Adverse Drug Events (ADEs), and are used for Pharmacovigilance (PhV), namely, drug safety surveillance. Exploration of analytical methodologies to enhance SRS-based discovery will contribute to more effective PhV. In this study, we proposed a statistical modeling approach for SRS data to address heterogeneity by a reporting time point. Furthermore, we applied this approach to analyze ADEs of incretin-based drugs such as DPP-4 inhibitors and GLP-1 receptor agonists, which are widely used to treat type 2 diabetes. **Methods:** SRS data were obtained from the Japanese Adverse Drug Event Report (JADER) database. Reported adverse events were classified according to the MedDRA High Level Terms (HLTs). A mixed effects logistic regression model was used to analyze the occurrence of each HLT. The model treated DPP-4 inhibitors, GLP-1 receptor agonists, hypoglycemic drugs, concomitant suspected drugs, age, and sex as fixed effects, while the quarterly period of reporting was treated as a random effect. Before application of the model, Fisher's exact tests were performed for all drug-HLT combinations. Mixed effects logistic regressions were performed for the HLTs that were found to be associated with incretin-based drugs. Statistical significance was determined by a two-sided p-value <0.01 or a 99% two-sided confidence interval. Finally, the models with and without the random effect were compared based on Akaike's Information Criteria (AIC), in which a model with a smaller AIC was considered satisfactory.
**Results:** The analysis included 187,181 cases reported from January 2010 to March 2015. It showed that 33 HLTs, including pancreatic, gastrointestinal, and cholecystic events, were significantly associated with DPP-4 inhibitors or GLP-1 receptor agonists. In the AIC comparison, half of the HLTs reported with incretin-based drugs favored the random effect, whereas HLTs reported frequently tended to favor the mixed model. **Conclusion:** The model with the random effect was appropriate for analyzing frequently reported ADEs; however, further exploration is required to improve the model. The core concept of the model is to introduce a random effect of time. Modeling the random effect of time is widely applicable to various SRS data and will improve future SRS data analyses.

## INTRODUCTION

The incretins are a group of intestinal hormones that stimulate insulin secretion. During the last decade, several hypoglycemic drugs based on incretin have gained widespread use as treatments for patients with type 2 diabetes. Incretin-based drugs are classified as inhibitors of incretin-degrading protease Dipeptidyl Peptidase 4 (DPP-4) or as incretin hormone Glucagon-Like Peptide 1 (GLP-1) receptor agonists. DPP-4 inhibitors and GLP-1 receptor agonists lower fasting and postprandial glucose, but do not produce hypoglycemia and are not associated with body weight gain or reduced in blood pressure (*Nauck, 2013*).

DPP-4 inhibitors and GLP-1 receptor agonists have been associated with adverse outcomes, including pancreatic disorders, although some of these findings are controversial (*Butler et al., 2013*; *Devaraj & Maitra, 2014*; *Nauck, 2013*). An analysis of the US Food and Drug Administration (FDA) Adverse Event Reporting System (FAERS) revealed that use of DPP-4 inhibitor sitagliptin or GLP-1 receptor agonist exenatide increased the Odds Ratio (OR) for pancreatitis more than 6-fold, while increasing the OR for pancreatic cancer more than 2-fold, in comparison with other medications (*Elashoff et al., 2011*); however, most other clinical studies have demonstrated no evidence suggesting such risks (*Li et al., 2014*; *Nauck, 2013*). The FDA and European Medicines Agency (EMA) explored multiple streams of data and agreed that assertions concerning a causal association between incretin-based drugs and pancreatitis or pancreatic cancer were inconsistent with the current data (*Egan et al., 2014*).

Spontaneous Reporting Systems (SRSs) such as the FAERS are passive systems composed of reports of suspected Adverse Drug Events (ADEs) collected from healthcare professionals, consumers, and pharmaceutical companies (*Harpaz et al., 2012*). SRSs play an essential role in Pharmacovigilance (PhV), which is also referred to as drug safety surveillance. Although SRSs cover large populations, their data have some biases in reporting. In the case of incretin-based drugs in the FAERS, reporting of pancreatitis was largely influenced by the relevant FDA alerts in an example of notoriety bias, which could cause overestimation of risk (*Raschi et al., 2013*). Spontaneous reporting patterns change over time, which is one of the limitations of SRSs (*Bate & Evans, 2009*). SRSs have numerous limitations; nevertheless, PhV has relied predominantly on SRSs (*Gibbons et al., 2010*; *Harpaz et al., 2012*). Therefore, exploration of novel analytical methodologies to enhance SRS-based discovery will highlight the value of SRSs and contribute to more effective PhV.

The most conventional analytical approach for SRS data is Disproportionality Analyses (DPA). DPA methodologies use frequency analysis of $2 \times 2$ contingency tables to quantify the degree to which a drug-event combination co-occurs disproportionately in comparison with that expected in the absence of an association (*Bate & Evans, 2009*; *Harpaz et al., 2013*). DPA do not require complicated modeling;

however, these do not adjust for confounding factors. The alternatives to DPA are multivariate modeling techniques such as logistic regression. Multivariate modeling techniques can adjust for potential confounding and masking factors during the analysis of drug-event relationships (*Harpaz et al., 2013*). In a performance evaluation using the FAERS, approaches based on logistic regression outperformed DPA approaches (*Harpaz et al., 2013*).

The objectives of the present study are to propose a novel multivariate modeling approach for SRS data and to apply this approach to analyze ADEs associated with incretin-based drugs from an SRS. We designed a mixed effects logistic regression model and performed comprehensive analyses using this model. The analyzed data were obtained from the Japanese Adverse Drug Event Report (JADER) database maintained by the Pharmaceuticals and Medical Devices Agency (PMDA) (*PMDA, 2015*). Most case reports in the FAERS are from consumers or lawyers, whereas those in the JADER are medically confirmed (*Nomura et al., 2015*). The analyses were based mainly on mixed effects logistic regression, in which DPA were used adjunctively. Mixed effects logistic regression models contain variables for random effects in addition to those for fixed effects similar to conventional logistic regression models. The random variable in a logistic regression model describes the ramifications of different sources of heterogeneity and associations between outcomes (*Larsen et al., 2000*). Mixed effects logistic regression model is a type of Generalized Linear Mixed Models (GLMMs). The use of GLMMs in medical literature has recently increased to take into account data correlations when modeling binary or count data (*Casals, Girabent-Farres & Carrasco, 2014*); however, applications of GLMMs to SRS data have been reported rarely. As a rare application of GLMMs to SRS data, an approach based on a mixed effects Poisson regression model was proposed (*Gibbons et al., 2008*). This approach yields rate multipliers for each drug in a class of drugs, which describe the deviation of the rate for a specific adverse event from that for the drug class as a whole. In contrast, the present approach is based on a logistic regression model with a random effect of time and addresses heterogeneity between time points. Since reporting-time-point is an attribute common to SRSs, modeling the random effect of time is widely applicable to any ADE in any SRS. Nevertheless, to the best of our knowledge, modeling approaches such as this have not been reported. This is the first application of a logistic regression model with a random effect of time to SRS data.

## MATERIAL AND METHODS

### Data source

The JADER dataset, which was published in July 2015 and contained 353,988 unique cases, was obtained from the website of the PMDA. The analyzed cases were reported from January 2010 to March 2015 and had available records regarding age and sex.

Adverse events in the JADER were coded as "Preferred Terms" (PTs) in the Japanese version of the Medical Dictionary for Regulatory Activities (MedDRA/J) (*MedDRA, 2015*). MedDRA has a hierarchical structure, in which PTs are grouped into "High Level Terms" (HLTs), HLTs are grouped into "High Level Group Terms" (HLGTs),

and HLGTs are grouped into "System Organ Classes" (SOCs). Before data analyses, a relational database was constructed from the JADER dataset and MedDRA/J version 18.0. SQLite version 3.8.5 was used as the database management system (*SQLite Development Team, 2015*).

As incretin-based drugs, all DPP-4 inhibitors and GLP-1 receptor agonists approved in Japan were assessed. The DPP-4 inhibitors were sitagliptin phosphate hydrate, vildagliptin, alogliptin benzoate, alogliptin benzoate/pioglitazone hydrochloride (combination drug), linagliptin, teneligliptin hydrobromide hydrate, anagliptin, and saxagliptin hydrate. The GLP-1 receptor agonists were exenatide, liraglutide, and lixisenatide.

## Data analysis

The analysis of ADEs associated with incretin-based drugs was composed of two phases. The first phase was a DPA based on Fisher's exact test. The second phase was a multivariate analysis using a mixed effects logistic regression model.

The PTs of ADEs were classified according to the MedDRA HLTs. All combinations of generic drug names and HLTs were extracted. Fisher's exact tests were performed for all combinations of incretin-based drugs and reported HLTs. Associations that yielded a two-sided p-value <0.01 and an OR >1 were considered significant.

Mixed effects logistic regressions were performed for each HLT significantly associated with incretin-based drugs. The mixed effects logistic regression model was as follows:

$$\frac{P(Y_i = 1|x_i, z_i)}{P(Y_i = 0|x_i, z_i)} = exp(x_i^T \beta + z_i^T u)$$

Where $Y_i$ is a binary variable describing the outcome of case $i$ (0 or 1), $\beta$ is a fixed parameter vector, $x_i$ is a covariate vector for fixed effects, $u$ is a vector of random variables from probability distributions, and $z_i$ is a covariate vector for random effects. $u$ represents unmeasured covariates as a way of modeling heterogeneity and correlated data (*Larsen et al., 2000*).

In the newly developed model, the binary outcome was whether or not each HLT was reported. For fixed effects, the covariates were use of DPP-4 inhibitors, use of GLP-1 receptor agonists, use of any hypoglycemic drugs (an alternative indicator for hyperglycemia), sum of concomitant suspected drugs (determined by reference to the Fisher's exact tests), age (in 10-year intervals), and sex. The random effect was the quarterly period of reporting. The variables for the random effect were random intercepts normally distributed with mean 0 and one common variance. The associations between incretin-based drugs and HLTs were assessed by ORs with 99% two-sided Wald-type Confidence Intervals (CIs). Because the present analysis was an exploratory screening, the problem of multiple comparison was not addressed. Instead of correcting that, stringent levels of statistical significance were set (p-value <0.01 and 99% CI).

The newly developed mixed model was compared with a fixed model that did not include the random effect. The covariates for fixed effects in the fixed model were the same covariates use in the mixed model. Logistic regressions based on each model were

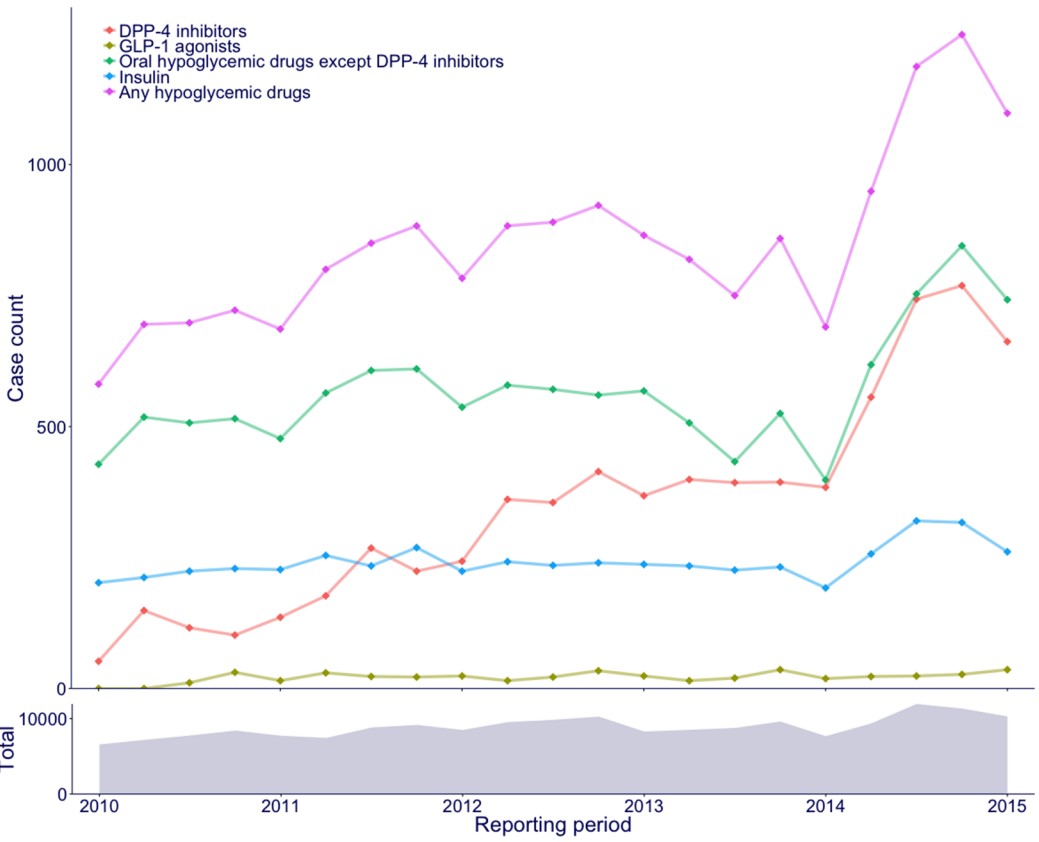

**Figure 1 Case counts of hypoglycemic drugs by each quarterly period.** The line plot denotes cases reported with hypoglycemic drugs. The area plot denotes all cases.

performed for all reported HLTs associated with incretin-based drugs. Subsequently, the adequacy of the model was assessed by Akaike's Information Criteria (AIC) (*Burnham & Anderson, 2002*). A model with a smaller AIC was favored.

All analyses were performed using the R version 3.2.1 (*R Development Core Team, 2010*). The glmmML package version 1.0 was used with the "ghq" (Gauss-Hermite quadrature) method for the mixed effects logistic regressions (*Broström, 2013*).

## RESULTS

### Description of the analyzed case reports

The JADER included 204,472 unique cases that were reported from January 2010 to March 2015, of which 187,181 had available records for age and sex and were analyzed. The records included 4,952 generic drug names and 6,151 PTs under 1,377 HLTs. DPP-4 inhibitors were mentioned in 7,265 cases, whereas GLP-1 receptor agonists were mentioned in 451 cases. Figure 1 shows the number of cases mentioning hypoglycemic drugs that were reported during each quarterly period. Although the number of cases for other hypoglycemic drugs increased gradually over time, the number of cases for DPP-4 inhibitors increased markedly.

**Table 1 Case counts of adverse events associated with DPP-4 inhibitors or GLP-1 receptor agonists.**

| MedDRA HLT | DPP-4 inhibitors | GLP-1 receptor agonists | Total |
|---|---|---|---|
| Thyroid neoplasms | 1 | 3 | 62 |
| Thyroid neoplasms malignant | 0 | 2 | 53 |
| Cystic pancreatic disorders | 2 | 1 | 16 |
| Pancreatic disorders NEC | 11 | 3 | 50 |
| Adrenal cortical hypofunctions | 5 | 4 | 184 |
| Gastrointestinal neoplasms benign NEC | 5 | 2 | 22 |
| Chronic polyneuropathies | 3 | 2 | 44 |
| Pancreatic neoplasms | 47 | 16 | 166 |
| Cholecystitis and cholelithiasis | 39 | 12 | 441 |
| Bile duct infections and inflammations | 9 | 4 | 176 |
| Pancreatic neoplasms malignant (excl islet cell and carcinoid) | 42 | 13 | 142 |
| Injection site reactions | 6 | 8 | 742 |
| Non-mechanical ileus | 16 | 7 | 325 |
| Diabetic complications NEC | 23 | 19 | 177 |
| Acute and chronic pancreatitis | 234 | 29 | 1038 |
| Gastrointestinal atonic and hypomotility disorders NEC | 20 | 9 | 390 |
| Gastric neoplasms malignant | 19 | 5 | 279 |
| Benign neoplasms gastrointestinal (excl oral cavity) | 16 | 3 | 95 |
| Skin autoimmune disorders NEC | 27 | 0 | 186 |
| Rheumatoid arthropathies | 17 | 1 | 153 |
| Rheumatoid arthritis and associated conditions | 17 | 1 | 154 |
| Hyperglycaemic conditions NEC | 92 | 34 | 728 |
| Arthropathies NEC | 24 | 0 | 417 |
| Lower respiratory tract neoplasms | 26 | 4 | 393 |
| Lower gastrointestinal neoplasms benign | 10 | 2 | 51 |
| Diabetic complications neurological | 15 | 4 | 71 |
| Gastrointestinal stenosis and obstruction NEC | 114 | 11 | 1216 |
| Urinalysis NEC | 36 | 1 | 149 |
| Digestive enzymes | 23 | 2 | 249 |
| Metabolic acidoses (excl diabetic acidoses) | 98 | 14 | 611 |
| Skeletal and cardiac muscle analyses | 66 | 1 | 896 |
| Non-site specific injuries NEC | 76 | 4 | 1179 |
| Coronary necrosis and vascular insufficiency | 141 | 12 | 1555 |

## Mixed effects logistic regressions

The cases associated with incretin-based drugs included 1,430 PTs under 735 HLTs. The Fisher's exact tests showed that 106 of the 735 HLTs were significantly associated with any incretin-based drug (two-sided p-value <0.01 and OR >1). In the mixed effects logistic regressions, 33 of the 106 HLTs identified by the Fisher's exact tests were significantly associated with DPP-4 inhibitors or GLP-1 receptor agonists (99% two-sided CI). Table 1 shows the number of cases reported for each HLT.

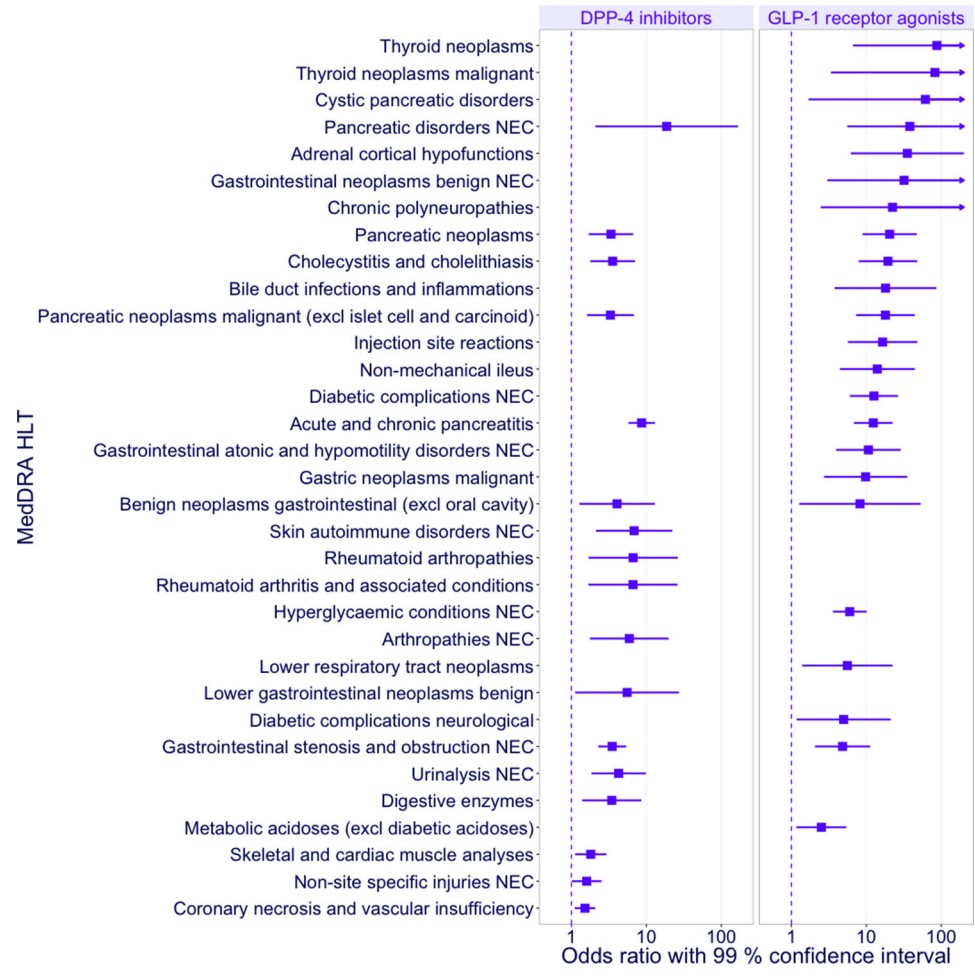

**Figure 2 Odds ratios of the adverse events associated with DPP-4 inhibitors or GLP-1 receptor agonists.** The forest plot denotes odds ratios (ORs) with 99% confidence intervals (CIs) for each event. Significant ORs with CIs are plotted.

Figure 2 shows ORs with 99% CIs for the significant associations between HLTs and DPP-4 inhibitors or GLP-1 receptor agonists. "NEC" in the MedDRA terms is an acronym for "Not Elsewhere Classified," which denotes groupings of miscellaneous terms, whereas "excl" is an abbreviation of "excluding." The HLTs associated with DPP-4 inhibitors included "Pancreatic disorders NEC" (OR 18.66; 99% CI 2.09–166.25) and "Acute and chronic pancreatitis" (8.65; 5.76–12.98). The HLTs associated with GLP-1 receptor agonists included "Thyroid neoplasms" (87.25; 6.64–1146.27) and "Cystic pancreatic disorders" (61.32; 1.69–2224.49). The HLTs associated with DPP-4 inhibitors and GLP-1 receptor agonists indicated pancreatic events ("Acute and chronic pancreatitis," "Pancreatic neoplasms," "Pancreatic neoplasms malignant (excl islet cell and carcinoid)," and "Pancreatic disorders NEC"), gastrointestinal events ("Benign neoplasms gastrointestinal (excl oral cavity)" and "Gastrointestinal stenosis and obstruction NEC"), and cholecystic events ("Cholecystitis and cholelithiasis"). Although DPP-4 inhibitors and GLP-1 receptor agonists were not associated with

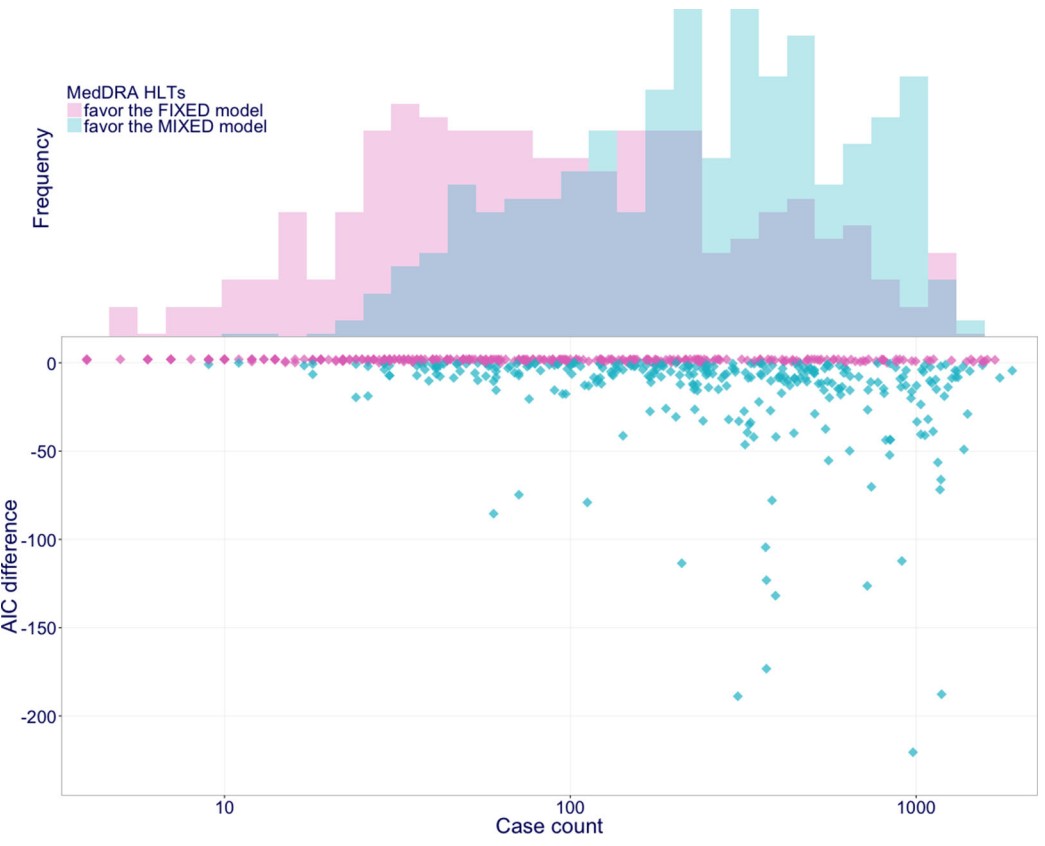

**Figure 3 AIC improvements with a random effect.** The vertical axis of the scatter plot denotes AIC differences calculated by subtracting that of the fixed model from that of the mixed model. When the AIC difference is less than 0, the mixed model is favored. The horizontal axis denotes total case counts for each MedDRA HLT. The histogram denotes their frequencies.

hypoglycemic events, GLP-1 receptor agonists were associated with several HLTs related to diabetes, including "Hyperglycaemic conditions NEC" and "Diabetic complications NEC."

## Comparison between the models with and without the random effect

Figure 3 shows the comparison between the models with (mixed model) and without (fixed model) the random effect. In 604 of the 735 HLTs reported with incretin-based drugs, the AIC of the models were calculated normally. Of the 604 HLTs, 302 favored the mixed model, whereas the others favored the fixed model. The median number of reported cases for the group of HLTs favoring the mixed model was 264, whereas the median number of reported cases for the group of HLTs favoring the fixed model was 83; thus, HLTs reported frequently tended to favor the mixed model.

## DISCUSSION

### Time-series variation of spontaneous reports

SRSs accumulate a large amount of data regarding ADEs every year; thus, the contents of SRSs are not constant. In the present study, the report composition of

hypoglycemic drug groups varied during the study period. Reports associated with DPP-4 inhibitors showed a marked increase in comparison with that of reports associated with other hypoglycemic drugs, perhaps because of an increasing number of approved products and an associated increase in drug use. The number of reports during a particular period is affected by numerous factors. This variation in reporting results in temporal heterogeneity, supporting the appropriateness of the mixed model.

### Adverse events associated with incretin-based drugs

Some HLTs associated with incretin-based drugs in the present study have been reported as issues of concern in previous studies. Some groups of similar HLTs, e.g., "Thyroid neoplasms" and "Thyroid neoplasms malignant" were identified because some PTs are linked to multiple HLTs in the MedDRA. In comparison with DPP-4 inhibitors, GLP-1 receptor agonists showed relatively wide CIs for some of the HLTs because fewer cases were reported, leading to unreliable results.

In the present study, pancreatic disorders, including pancreatitis and pancreatic cancer, were associated with DPP-4 inhibitors and GLP-1 receptor agonists, which were consistent with results obtained via analysis of the FAERS (*Butler et al., 2013*; *Elashoff et al., 2011*). In addition, thyroid cancer was associated with GLP-1 receptor agonists; however, because of the small number of cases, this finding is unreliable. Analyses of the FAERS also indicated that GLP-1 receptor agonists increased ORs for thyroid cancer (*Butler et al., 2013*; *Elashoff et al., 2011*). Thyroid cancer and pancreatic disorders are among the most controversial safety concerns regarding incretin-based drugs; however, no evidence has been found for such risks in human clinical studies (*Butler et al., 2013*; *Nauck, 2013*). The other HLTs associated with DPP-4 inhibitors and GLP-1 receptor agonists were "Benign neoplasms gastrointestinal (excl oral cavity)," "Gastrointestinal stenosis and obstruction NEC," and "Cholecystitis and cholelithiasis." Gastrointestinal events such as nausea, vomiting, and diarrhea are common ADEs of incretin-based drugs (*Nauck, 2011*); however, benign gastrointestinal neoplasms, stenosis, and obstruction have not been referred to in past studies. Cholecystic events have not in the same way.

Hypoglycemia, an adverse event associated with some hypoglycemic drugs, was not associated with incretin-based drugs. In contrast, hyperglycemia and several other diabetic complications were associated with GLP-1 receptor agonists, perhaps because of cases of ineffective drug treatment.

### Limitations

The limitations of SRS data mining include confounding by indication (i.e., patients taking a particular drug may have a disease that is itself associated with a higher incidence of the adverse event), systematic under-reporting, questionable representativeness of patients, effects of media publicity on numbers of reports, extreme duplication of reports, and attribution of the event to a single drug when patients may be exposed to multiple drugs (*Gibbons et al., 2010*). In addition, spontaneous reports do not reliably detect adverse drug reactions that occur widely separated in time from the original use of the drug (*Brewer & Colditz, 1999*).

The newly developed model reported here addresses the confounding influences of temporal heterogeneity and concomitant drug use. Nevertheless, risks identified via analysis of SRS data should be considered as safety signals, rather than definitive statements of cause and effect. For further interpretation of each ADE, additional reviews of other data sources are recommended.

## Mixed effects logistic regression model

In the AIC comparison between the mixed model and the fixed model, half of the HLTs reported with incretin-based drugs favored each model. The HLTs that favored the mixed model were reported more frequently than those that favored the fixed model, indicating that the mixed model may be more appropriate for frequently reported ADEs. The AIC formula has a bias-correction term for the number of estimable parameters (*Burnham & Anderson, 2002*). In the above comparison, the mixed model has only one more parameter than does the fixed model; hence, the difference between the penalties for the correction is small.

The adequacy of the random effect was demonstrated; however, the model can be improved. Although we assumed a normal distribution for the random effect, the appropriateness of this assumption is unclear. Moreover, it is unclear whether a single probability distribution is sufficient to assess the random effect on the widely spread time-scale of spontaneous reports. Sampling of parameter distributions by Bayesian hierarchical modeling is a potential solution to these problems. Currently, diverse implementations of Bayesian methods, which support practice of such modeling, are accessible (*Li et al., 2011*; *MacLehose & Hamra, 2014*).

## CONCLUSION

We proposed a logistic regression model for SRS data taking into account the random effect of time and applied this model to analyze ADEs reported with incretin-based drugs in the JADER. The newly developed model was appropriate for ADEs reported frequently; however, further exploration is required to improve the model.

The core concept of the newly developed model is to introduce a random effect of time. The random effect introduced into a univariate or multivariate model addresses heterogeneity by time unit. Reporting-time-point is an attribute common to all spontaneous reports. Hence, modeling the random effect of time is widely applicable to various SRS data and will improve future statistical analyses of SRS data.

### Funding

This work was supported by a grant from the Japanese Ministry of Health, Labour and Welfare. The funders had no role in study design, data collection and analysis, decision to publish, or preparation of the manuscript.

### Competing Interests

The authors declare that they have no competing interests.

## Author Contributions

- Daichi Narushima conceived and designed the experiments, performed the experiments, analyzed the data, contributed reagents/materials/analysis tools, wrote the paper, prepared figures and/or tables, reviewed drafts of the paper.
- Yohei Kawasaki analyzed the data, wrote the paper, prepared figures and/or tables, reviewed drafts of the paper.
- Shoji Takamatsu conceived and designed the experiments, contributed reagents/materials/analysis tools, reviewed drafts of the paper.
- Hiroshi Yamada conceived and designed the experiments, reviewed drafts of the paper.

## Data Deposition

The Japanese authority, PMDA, which owns this date, does not permit sharing of the data directly. It can be accessed here: https://www.pmda.go.jp/safety/info-services/drugs/adr-info/suspected-adr/0005.html (in Japanese).

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
