# Peer review of "Adverse events associated with incretin-based drugs in Japanese spontaneous reports: a mixed effects logistic regression model"

_PeerJ, doi:10.7717/peerj.1753_

## Round 0.1 · original submission · Minor Revisions

Dear authors,

First of all, please let me apologise for the delay in handling this manuscript: it has been quite difficult to recruit referees, probably because of the new year holidays.

Beside the comments provided by the Referee, please find below some additional comments of my own.

While I find the manuscript technically sound, I also find it very technical, with many items that are not discussed in detail; considering the multi-disciplinary readership of PeerJ, this may alienate many potential readers - which would, of course, be bad for the diffusion of your work.

Some more specific issues:

Introduction
While the introduction is generally OK, I find that it lacks a more detailed review of the available literature. Many papers have been published in the past about FAERS, for instance; but very few of them are cited.
Moreover, most of the references of the introduction are focused on the problems associated with incretin drugs, while the conclusions are completely focused on the methodology. Authors should find a better equilibrium between both aspects: if the methodology is new, this should be demonstrated by correctly citing and describing relevant works.

Material and Methods
The authors should take into account that PeerJ readership come from very different disciplines, and therefore any paper should include clear explanations of main terms and concepts. Specifically, I find that too many acronyms are used (for instance, in the Results section); and that many of them are not clearly explained. For instance, the average reader may have no idea about what a Preferred Term or a High Level Term is!

Methods
The Method section should be extended, and include more information about the type of statistical analysis used. Once again, authors should bear in mind that the average reader may not have a strong statistical background. As noted by the Referee, there is no information about how the AIC should be used and interpreted.

Conclusions
Just three lines of conclusions are far from enough. I'd recommend to present a more exhaustive review of the results, organised in two main lines: the advantages of the proposed methodology on one hand; and the specific results for incretin-based drugs on the other.
I think that this last point is especially relevant: a more detailed discussion should be included about the fact that some drugs are more dangerous than others, and compare this with the existing literature.

Reviewer 1 ·

Basic reporting

The authors should pay particular attention to the instructions given by the Author Guidelines and revise the references especially in body text citations accordingly. For example, Page3. line81: [10], Page4. line104: [15], and line106: [16].

Experimental design

1.It would be more reader-friendly if the authors give more information about how to judge a favor model using the AIC. Did the authors determine a favor model with a minimal AIC or the difference of AIC in two models was negative?
2.Why did the authors use 99% two-side (‘two-side’ should be written in body text) confidence intervals (CIs) instead of 95%? Was it required stringent level of statistical significance to evaluate a favor model? I think it would be useful if the authors give more information about it.

Validity of the findings

1. In the results of odds ratios of mixed effects logistic regression, some of the CIs were very wide. What influence did the wide CIs have on the model fit, or no impact? The authors’ view or opinions should be describe in the discussion section as necessary.
2. The generalizability of this study results is questionable. Is the approach only for SRS data from JADER or will it expand for the other data or database?

Additional comments

The paper by Daichi Narushima et al. entitled ‘Adverse Events Associated with Incretin-based Drugs in Japanese Spontaneous Reports: A Mixed Effects Logistic Regression Model’ is an interesting study to proposed a statistical modeling approach based on a logistic regression model to address heterogeneity by time of reporting for Spontaneous reporting systems (SRSs) data in the Japanese Adverse Drug Event Report (JADER) database. The authors applied this approach to analyze ADEs of incretin-based drugs. The authors reported that the model with the random effect was appropriate for frequently reported adverse drug events (ADEs). This was a novel approach and the findings will be of interest to medical experts as well as public officials and researchers in this field. However, I recommend several revision points.

---

## Round 0.2 · accepted · Accept

I would like to thank the authors for having addressed the comments highlighted in the previous evaluation round. The paper is now more complete, and ready to be published.

Reviewer 1 ·

Basic reporting

No Comments.

Experimental design

No Comments.

Validity of the findings

No Comments.

Additional comments

The manuscript is now successfully revised and acceptable in this journal.